# Efficient Continue Training of Temporal Language Model with Structural Information

**Zhaochen Su**[1], **Juntao Li**[1]*, **Zikang Zhang**[1], **Zihan Zhou**[2], **Min Zhang**[1]

[1]Institute of Computer Science and Technology, Soochow University, China
[2]Department of Chinese Language and Literature, Peking University
{suzhaochen0110,zikangzhangXQ}@gmail.com;
zhouzihan@pku.edu.cn; {ljt,minzhang}@suda.edu.cn

## Abstract

Current language models are mainly trained on snap-shots of data gathered at a particular time, which decreases their capability to generalize over time and model language change. To model the *time* variable, existing works have explored temporal language models (e.g., TempoBERT) by directly incorporating the timestamp into the training process. While effective to some extent, these methods are limited by the superficial temporal information brought by timestamps, which fails to learn the inherent changes of linguistic components. In this paper, we empirically confirm that the performance of pre-trained language models (PLMs) is closely affiliated with syntactically changed tokens. Based on this observation, we propose a simple yet effective method named *Syntax-Guided Temporal Language Model* (SG-TLM), which could learn the inherent language changes by capturing an intrinsic relationship between the *time* prefix and the tokens with salient syntactic change. Experiments on two datasets and three tasks demonstrate that our model outperforms existing temporal language models in both memorization and generalization capabilities. Extensive results further confirm the effectiveness of our approach across different model frameworks, including both encoder-only and decoder-only models (e.g., LLaMA). Our code is available at https://github.com/zhaochen0110/TempoLM.

## 1 Introduction

While Pre-trained Language Models (PLMs) have achieved remarkable success on most of NLP tasks (Qiu et al., 2020), they neglect the *time variable* due to their training on snap-shots of data collected at a particular point of time (Devlin et al., 2019; Liu et al., 2019). However, our language is constantly evolving and changing with new words being created (Huang et al., 2014; Rudolph and Blei, 2018; Amba Hombaiah

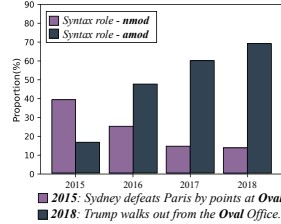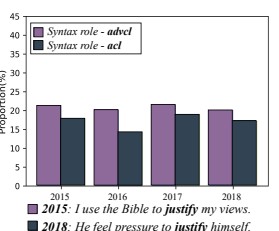

Figure 1: Representative examples of syntactic role changes across time. We track the two most frequent syntactic roles of the word *"oval"* and *"justify"* from 2015 to 2018, where the syntactic role distribution of the word *"justify"* is stable across years, while the word *"oval"* varies dramatically at different times.

et al., 2021) and existing words changing in meanings and usage (Labov, 2011; Eisenstein et al., 2014; Giulianelli et al., 2020; Montanelli and Periti, 2023). This *"static"* training paradigm prevents PLMs from generalizing over time and modeling language change (Lazaridou et al., 2021; Søgaard et al., 2021; Loureiro et al., 2022).

To make the model training more *"dynamic"*, existing studies have explored temporal language models (TLMs), which model temporality by incorporating timestamp directly into representation when pre-training LMs, i.e., TempoBERT (Rosin et al., 2022), TempoT5 (Dhingra et al., 2022). Their methods involve prepending a special time token to each sequence in the training data. Through stacking multiple attention layers, each token at different positions can capture the temporal information brought by time. With different time prefixes, PLMs can adaptively compute the corresponding temporal representations. Moreover, TLMs have better generalization[1] capability over time as standard LMs are unaware of which data is *"new"* and which is *"old"* due to the absence of timestamps during the training process (Dhingra et al., 2022).

Though the methods mentioned above can cap-

---

*Juntao Li is the Corresponding Author.

[1]Temporal generalization (Lazaridou et al., 2021) refers to a model is trained on data before time $T$ but is tested after $T$.

ture temporal information to a certain degree, they can merely incorporate "superficial" temporal information provided by the time prefix with the Masked Language Model (MLM) objective. Thus, it is natural to leverage more temporal-specific information captured by textual tokens, e.g., utilizing a small set of lexicons with salient lexical semantic change (Hamilton et al., 2016; Giulianelli et al., 2020; Tahmasebia et al., 2021) in the very recent LMLM method (Su et al., 2022). In this paper, we launch a thorough study to explore the effects of different methods for lexicon selection based on statistical patterns or linguistic attributes and find that the distributional change of syntactic roles (Kutuzov et al., 2021) is the most effective strategy in temporal-specific lexicon selection. As shown in Figure 1, the discrete syntactic role distribution of the word *"oval"* dramatically changes over time, while the word *"justify"* is stable across years.

Based on the above observations, we propose a *Syntax-Guided Temporal Language Model* (SG-TLM), which consists of two masking strategies: a *Syntax-Guided Masking* (SGM) and a *Temporal-Aware Masking* (TAM) strategy. Experimental results demonstrate that our proposed method significantly improves performance over other TLMs methods on two datasets and three tasks. Extensive results further confirm the efficiency of our method than the state-of-the-art lexicon selection solution based on semantic change, remarkable transferability across various model frameworks, and its positive impact on adaption to future data.

**Summary of Contributions:** (i) We explore the task of efficient syntax-guided lexicon selection, which is more challenging for static PLMs to predict on the time-stratified data. (ii) We propose a simple yet effective **S**yntax-**G**uided **T**emporal **L**anguage **M**odel (SG-TLM). (iii) SG-TLM exhibits excellent performance than other TLMs in terms of memorization and generalization for downstream tasks. (iv) Our method demonstrates superior efficiency than the SOTA solution, exhibiting high transferability across different model frameworks and positive adaptive ability to future data.

## 2 Preliminary Study

### 2.1 Temporal Language Model

Previous works have explored temporal language models to enhance the capability of PLMs in modeling language change and generalizing over time.

One popular method is to prepend a timestamp in different forms to a textual sequence, e.g., "`<2015>` Sydney defeats Paris by points at Oval." (Rosin et al., 2022), "`year: 2015` text: Sydney defeat Paris by points at Oval." (Dhingra et al., 2022), and utilize MLM objective to capture the temporal information brought by the time prefix. Through interacting each time prefix with the correlated textual tokens equally, temporal information can be injected into the pre-trained representation, which ignores the diachronic change degree of different tokens, e.g., time-specific tokens and time-agnostic tokens[2]. Thus, it is natural to enhance existing temporal language models by "accurately" injecting time information into these time-specific tokens, i.e., the core of building better temporal language models is to select these tokens with the time attribute. Normally, it is difficult to directly compute or estimate the diachronic change degree or the time attribute of tokens. Existing works mainly leverage the discrepancy of data across different periods to approximate the diachronic change degree of tokens. For instance, Su et al. (2022) measures the statistical distance of token representation across years to select these tokens (i.e., a lexicon) with salient lexical semantic change. Though effective, existing semantic-based lexicon selection methods require forwarding all training data with a large-scale language model in the data process stage and neglect structure information within the language. To accelerate the lexicon selection process and leverage structure information within languages, we explore the potential of syntactic role changes of different tokens in this paper that may benefit from the speed superiority of various syntactic parsing tools. We elaborate on more details of syntax-based lexicon selection below.

### 2.2 Syntax-Guided Lexicon Selection

We construct "syntax-guided lexicons" based on diachronic differences in syntax. According to Su et al. (2022), we first adopt YAKE! (Campos et al., 2018), a feature-based and unsupervised system to extract the candidate keywords $\mathcal{W}^t = \{w_1^t, w_2^t, \cdots, w_k^t\}$ from the texts $\mathcal{D}^t$ of time $t$. Then, we utilize off-the-shelf Stanza[3] (Manning et al., 2014) to automatically parse the syntax information for each sentence in the texts $\mathcal{D}^t$ and

---

[2]Time-specific tokens refer to these with salient (larger than a threshold) lexical semantic change, while the left tokens treated as time-agnostic ones in Su et al. (2022).

[3]https://github.com/stanfordnlp/stanza

count the frequency of syntactic roles for each candidate word $w_i$ and store them in the set $\mathcal{R}^t = (r_1^t, r_2^t, \cdots, r_i^t, \cdots, r_N^t)$. This set is a collection of dictionaries, with each dictionary representing the syntactic roles and their frequencies, which is structured as follows: $r_i^t = \{k_j^t : v_j^t\}_{j=1}^{|r_i^t|}$, where $k_j^t$ represents the syntactic role for word $w_i$ in time $t$, and $v_j^t$ is its frequency. For example, if the candidate word *"oval"* has the syntactic roles *"amod"* and *"nmod"* in time $t$ with frequencies 150 and 100, respectively, the corresponding dictionary in $\mathcal{R}^t$ would be $r_{oval}^t = \{amod : 150, nmod : 100\}$.

Using these syntactic dictionaries, we create feature vectors $\vec{a}_t$ and $\vec{a}_{t'}$ to represent the syntactic profiles of the candidate words in different periods. The size of the feature vectors $\vec{a}_t$ and $\vec{a}_{t'}$ may vary across words since we create separate feature lists for each word, including the corresponding syntactic roles. To align the vectors for each time, we pad the vectors with 0 for any missing syntactic roles. Finally, we calculate the cosine distance between $\vec{a}_t$ and $\vec{a}_{t'}$ to measure the difference between the syntactic profiles of the candidate words $\mathcal{W}^t$. We use a hyper-parameter $k$ to control the degree of syntactic change, ranking the candidate words $\mathcal{W}^t$ based on their cosine values and selecting the top-$k$ words as the syntax-guided lexicon, which consists of the tokens with significantly changed syntactic roles across different periods. Our proposed lexicon selection method is much faster than those used in LMLM (Su et al., 2022), in which their computation cost will be discussed in Section 4.4.

## 2.3 Discussions and Observations

In Section 2.2, we propose a direct and efficient syntax-guided approach for obtaining the lexicons which have undergone significant syntactic change over time. Following Su et al. (2022), we mask the tokens in the selected lexicons and utilize perplexity (*ppl.*) as a qualitative measure to compare the influence of different lexicon selection methods on static PLMs. To complete this, we build a time-stratified corpus from publicly released crawl news[4] datasets, which contains 1M English news articles for each year between 2014 and 2018. We post-tune the BERT[5] model with the data from 2014[6] and evaluate the four testing sets after 2015.

[4]https://data.statmt.org/news-crawl/
[5]We initialize parameters from BERT-BASE-UNCASED.
[6]To eliminate the impact of domain divergence (Gururangan et al., 2020) on the performance of testing, we post-tune the uniform BERT with the news data in 2014.

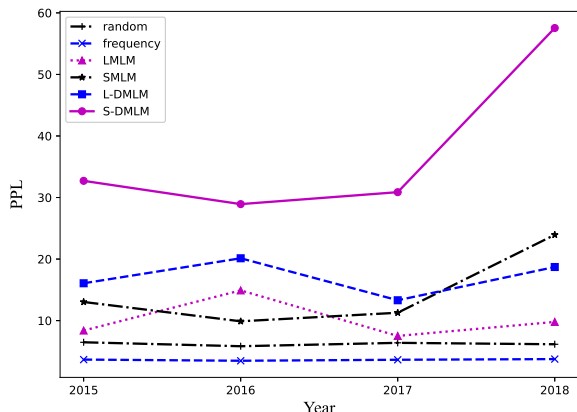

Figure 2: Results of the *ppl.* score. S-DMLM achieves the highest *ppl.* values among six selection methods.

**Methods for Comparison** We introduce six approaches to lexicon construction: (i) We first adopt two methods for extracting lexicons: random selection and frequency-aware selection. (ii) Besides, we introduce two approaches to selecting the lexicons with salient linguistic changes: lexical semantic change, i.e., LMLM (Su et al., 2022) and syntactic role change, i.e., our proposed syntax-guided method (SMLM) introduced in Section 2.2. (iii) Additionally, we consider words that are dependent on the tokens with significant diachronic change, as identified from the syntactic parsing process[7], i.e., the head node in the dependency parsing tree (McDonald et al., 2005), and propose two methods to include these words dependent on extracted lexicons based on LMLM and SMLM, named L-DMLM and S-DMLM, respectively.

**The Influence of Lexicon Selection Methods** The results are shown in Figure 2. We can see that: (i) In the absence of adding dependent words, the *ppl.* of SMLM is much higher than the other three lexicon selection methods, which indicates that it is more challenging for static PLMs to predict the lexicons selected from the syntactic perspective. (ii) After adding dependency information, both SMLM and LMLM methods show an apparent increase in their *ppl.* values, i.e., L-DMLM and S-DMLM, suggesting the positive impact of adding dependent words from syntactic parsing in lexicon selection. (iii) Above all, S-DMLM (marked with ●) achieves the highest *ppl.* values among six methods, which can select diachronic change lexicons that impose the most significant challenge to static LMs.

[7]This allows us to capture more nuanced and subtle temporal information present in the text, which is crucial for tasks that require temporal understanding and knowledge retention.

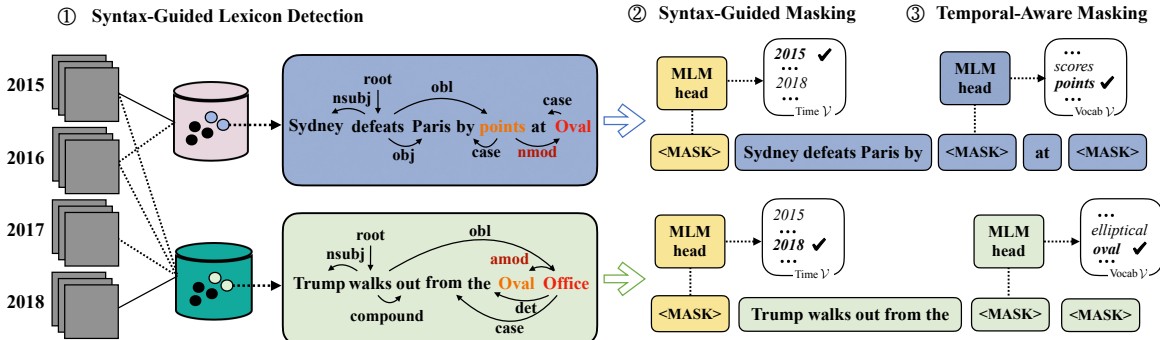

Figure 3: The overall structure of our proposed **S**yntax-**G**uided **T**emporal **L**anguage **M**odel (SG-TLM). We construct the syntax-guided lexicons across time (①) and post-train with two components: **S**yntax-**G**uided **M**asking (SGM) scheme (②) and **T**emporal-**A**ware **M**asking (TAM) scheme (③).

## 3 Syntax-Guided TLM

The Masked Language Model (MLM) objective (Devlin et al., 2019) is a widely-adopted self-supervised training method that involves randomly masking a certain percentage of the tokens in a text sequence and training a model to predict the masked tokens based on their context. Previous TLMs add a timestamp token at the beginning of the input sequence and utilize the MLM objective to predict the random masked tokens based on the context and the timestamp. However, these methods disregard inherent temporal-specific information provided by lexicons with salient change tokens. Based on the aforementioned observation[8], we propose *Syntax-Guided Temporal Language Model* (SG-TLM), which consists of two main components: a *Syntax-Guided Masking* (SGM) scheme and a *Temporal-Aware Masking* (TAM) method. Our proposed model is illustrated in Figure 3.

**Syntax-Guided Masking (SGM)**   We construct the syntax-guided[9] lexicons based on the distributional change of syntactic role across timestamps. Formally, given the text set $\mathcal{D}^t = \{d_1^t, d_2^t, \cdots, d_n^t\}$ at time $t$, we first rank the lexicons according to the word's cosine values of syntactic change. Then, we select $k$ ($k \in \{100, 200, \cdots, 500\}$) words with relative high scores as the masking candidate words $\mathcal{W}mask^t$. Considering the effectiveness of adding dependent words, we treat the words that are dependent by the candidate words $\mathcal{W}mask^t$ within

the sentence as additional temporal information. Specifically, given the masking ratio $\alpha^{10}$, we prioritize masking the words in $\mathcal{W}mask^t$ and their corresponding words in the dependency relationship[11]. We randomly mask the other tokens from the sequence if there are no sufficient masking candidates to meet the required number of masking tokens. Assuming it masks $m$ tokens in total and the sequence after masking at time $t$ is $d_i^{t'}$. The optimization objective of SGM can be written by:

$$\mathcal{L}_{\text{SGM}} = -\sum_{j=1}^{m} \log P(x = w_j | t, d_i^{t'}; \theta). \quad (1)$$

**Temporal-Aware Masking (TAM)**   Unlike previous work (Rosin et al., 2022), we predict masked tokens with salient syntactic role change and time tokens, given the remaining unmasked words within the sequence. Formally, given a sequence $\{d_i\}$ at time $t$, we denote $d_i^{(0)}$ for its timestamp and prepend the time token $t$ to $d_i$. Now the inputs to the model are a sequence $d_i = [t, d_i^{(1)}, d_i^{(2)}, d_i^{(3)}, \cdots, d_i^{(n)}]$. Assuming the set of masked token is $\{d_i^{(2)}\}$, we predict the time $t$ by the whole input text: $p(t) = p(t|d_i^{(1)}, \texttt{<MASK>}, d_i^{(3)}, \cdots, d_i^{(n)})$. As for the granularity of $t$, different values can be used according to the use case. In this work, we experiment with the granularity of a year for the **WMT** dataset, and a month for the **RTC** dataset.

---

[8]In Section 2.3, we discover that utilizing the distributional change of syntactic roles is the most effective strategy when selecting temporal-specific lexicons.

[9]In Appendix C, we will investigate the influence of parsing toolkit's performance on our SG-TLM.

[10]In Appendix 4.4, we will analyze suitable values for the masking ratio $a$ and word counts $k$.

[11]In Section 4.2, we also explore training the model on the tokens with salient syntactic change, without taking the corresponding dependent words (STLM) into account, and the results are shown in Table 1 (marked with †).

## 4 Experiments

### 4.1 Experimental Setup

**Datasets** We conduct continual pre-training on two datasets: WMT NEWS CRAWL (**WMT**) and REDDIT TIME CORPUS (**RTC**), respectively. The **WMT**[12] dataset, an open-domain dataset, consists of 4 million news articles published between 2015 and 2018: $\{\mathcal{D}_{2015}, \mathcal{D}_{2016}, \mathcal{D}_{2017}, \mathcal{D}_{2018}\}$. On the other hand, the **RTC** (Röttger and Pierrehumbert, 2021) is a monthly time-stratified dataset from March 2017 to February 2020. Unlike the **WMT** dataset, it specifically focuses on the political domain, enabling us to explore temporal dynamics within the specific domain. We select three months $\{\mathcal{D}_{2017-04}, \mathcal{D}_{2018-08}, \mathcal{D}_{2019-08}\}$ for pre-training, each containing 1 million unlabeled comments.

**Evaluation** Following the pre-training period, we evaluate the models' memorization (Dhingra et al., 2022) and generalization (Lazaridou et al., 2021) abilities by measuring their performance on downstream tasks[13]. To evaluate memorization, the model is tested on the same time steps as the pre-training data $\mathcal{S}_{1...T} = \{\mathcal{S}_1, \mathcal{S}_2, \ldots, \mathcal{S}_T\}$. To evaluate generalization, the model's performance is measured on future times ($\mathcal{S}_T \tilde{} \mathcal{S}_{T+n}$), which is invisible during the post-tuning stage.

After continual training on the **WMT** dataset, two tasks are used for model evaluation: political affiliation classification (**POLIAFF**) (Luu et al., 2022) and named entity recognition (**TWINER**) (Rijhwani and Preoţiuc-Pietro, 2020). The **POLIAFF** task involves fine-tuning the model with 10,000 labeled sentences from 2015. Testing includes $\{\mathcal{S}_{2015}, \mathcal{S}_{2016}, \mathcal{S}_{2017}, \mathcal{S}_{2018}\}$ for memorization and $\{\mathcal{S}_{2019}, \mathcal{S}_{2020}\}$ for generalization, with each year containing 2,000 specific sentences. For the **TWINER** task, 2,000 labeled tweets from 2015 are selected for fine-tuning. We evaluate memorization abilities using datasets $\{\mathcal{S}_{2016}, \mathcal{S}_{2017}, \mathcal{S}_{2018}\}$ and generalization capabilities using dataset $\{\mathcal{S}_{2019}\}$. Following pre-training on the **RTC** dataset, the model is evaluated on the political subreddit prediction (**PSP**) (Röttger and Pierrehumbert, 2021) task. Specifically, the model is fine-tuned on 20,000 labeled data samples from

April 2017, extracted from the same dataset used for pre-training. Memorization and generalization are tested on $\{\mathcal{S}_{2017-04}, \mathcal{S}_{2018-08}, \mathcal{S}_{2019-08}\}$ and $\{\mathcal{S}_{2020-01}, \mathcal{S}_{2020-02}\}$, respectively. We calculate the F1-score as the testing results for all our experiments. More details about the task and our model's training are shown in Appendix A.

### 4.2 Baselines

We establish several baselines that encapsulate different continual pre-training strategies. Firstly, we consider two naive baselines that do not incorporate timestamps during the pre-training stage: (i) **BERT (w/o)** (Devlin et al., 2019), which is directly fine-tuned on the downstream task without pre-training. (ii) **Uniform** involves training the model with mixed pre-trained data. Additionally, we adopt three up-to-date TLMs, which utilize timestamps during pre-training: (iii) **TAda** (Röttger and Pierrehumbert, 2021), which involves continual pre-training on specific time buckets to obtain separate models specialized for different periods. (iv) **Temporal** (Dhingra et al., 2022) integrates time $t$ as a prefix to the input during pre-training, with temporal-specific lexicons randomly generated. (v) **LMLM** (Su et al., 2022) is the SOTA of temporal adaptation, which strengthens PLMs' generalization with salient lexical semantic change. We utilize their method of lexicon construction. **STLM** and **SG-TLM** are our methods, with the distinction of whether to add dependent words during the lexicons construction. Appendix B offers additional training details on the compared TLMs.

### 4.3 Main Results

Table 1 presents the results on the **WMT** dataset. TLMs demonstrate superior performance compared to models that do not consider timestamps. However, previous TLMs such as **TAda** and **Temporal** show only marginal improvements over the Uniform model, indicating limited learned temporality information from the timestamps. Conversely, incorporating linguistic information into TLM training significantly improves both memorization and generalization. Among the evaluated baselines, SG-TLM achieves the highest average F1-scores on both datasets, i.e., 66.77 and 66.67, highlighting the effectiveness of leveraging syntax and dependency information within languages. Table 2 presents the results for the **RTC** dataset. Similar to the previous findings, SG-TLM consistently achieves the best performance.

---

[12]https://data.statmt.org/news-crawl/

[13]Previous works (Röttger and Pierrehumbert, 2021; Lazaridou et al., 2021) have discovered that the model's performance deteriorates as the gap between the training and testing time increases. Following their setting, we evaluate the model that was fine-tuned on the oldest time step.

| Model | POLIAFF | | | | | | | TWINER | | | | |
|---|---|---|---|---|---|---|---|---|---|---|---|---|
| | 2015 | 2016 | 2017 | 2018 | *2019* | *2020* | Avg. | 2016 | 2017 | 2018 | *2019* | Avg. |
| BERT | $91.30_{1.1}$ | $63.80_{0.6}$ | $52.22_{0.9}$ | $39.20_{0.6}$ | $45.00_{0.6}$ | $38.93_{1.0}$ | 55.08 | $75.82_{0.4}$ | $56.98_{0.2}$ | $56.16_{0.5}$ | $56.99_{0.4}$ | 61.48 |
| + Uniform | $91.33_{0.8}$ | $74.87_{0.5}$ | $66.01_{0.6}$ | $54.67_{0.7}$ | $55.75_{0.5}$ | $46.57_{0.9}$ | 64.87 | $77.04_{0.6}$ | $59.05_{0.4}$ | $58.52_{0.5}$ | $59.07_{0.3}$ | 63.42 |
| + TAda | $90.79_{0.9}$ | $74.40_{0.7}$ | $65.33_{0.4}$ | $55.23_{0.6}$ | $53.19_{0.4}$ | $44.95_{0.6}$ | 63.98 | $75.50_{0.4}$ | $60.69_{0.3}$ | $60.72_{0.4}$ | $60.12_{0.1}$ | 64.25 |
| + Temporal | $90.84_{0.6}$ | $73.46_{0.5}$ | $65.03_{0.8}$ | $55.39_{0.7}$ | $56.88_{0.2}$ | $46.41_{0.5}$ | 64.67 | $74.02_{0.5}$ | $57.54_{0.7}$ | $56.84_{0.3}$ | $57.54_{0.4}$ | 61.49 |
| + LMLM | $92.81_{0.7}$ | $75.82_{0.5}$ | $64.83_{0.3}$ | $55.46_{0.4}$ | $57.12_{0.5}$ | $48.20_{0.6}$ | 65.71 | $79.34_{0.3}$ | $60.75_{0.4}$ | $60.01_{0.6}$ | $60.76_{0.3}$ | 65.22 |
| + STLM † | $93.28_{0.5}$ | $75.56_{0.3}$ | $65.31_{0.4}$ | $55.71_{0.2}$ | $\mathbf{57.76_{0.4}}$ | $49.13_{0.2}$ | 66.13 | $81.67_{0.3}$ | $60.78_{0.2}$ | $60.09_{0.4}$ | $60.91_{0.3}$ | 65.86 |
| + SG-TLM ‡ | $\mathbf{93.69_{0.3}}$ | $\mathbf{76.69_{0.2}}$ | $\mathbf{66.58_{0.1}}$ | $\mathbf{55.94_{0.4}}$ | $57.41_{0.1}$ | $\mathbf{50.32_{0.2}}$ | **66.77** | $\mathbf{81.78_{0.2}}$ | $\mathbf{61.85_{0.3}}$ | $\mathbf{61.18_{0.1}}$ | $\mathbf{61.87_{0.2}}$ | **66.67** |

Table 1: Results of the **WMT** dataset: memorization and generalization performance on the **POLIAFF** and **TWINER** tasks. The timestamp represented in italics are not visible during the post-tuning stage, i.e., *2019* and *2020* in **POLIAFF** dataset and *2019* in **TWINER** dataset. We utilize the above testing sets to evaluate the models' generalization and the left to test the memorization. STLM (marked with †) and SG-TLM (marked with ‡) are our methods, the difference is whether to include dependent words during the masking scheme. Our proposed SG-TLM achieves the highest average F1-score on two tasks. Each number is the average of 5 runs with different seeds.

| Model | RTC | | | | |
|---|---|---|---|---|---|
| | 17-04 | 18-04 | 19-08 | *20-01* | *20-02* |
| BERT | $49.61_{0.8}$ | $41.74_{1.1}$ | $38.23_{0.5}$ | $38.84_{1.2}$ | $39.97_{0.3}$ |
| + Uniform | $51.97_{0.7}$ | $42.57_{1.3}$ | $39.98_{0.4}$ | $40.75_{0.9}$ | $40.53_{1.0}$ |
| + TAda | $50.62_{0.6}$ | $42.21_{0.9}$ | $38.43_{1.4}$ | $39.47_{0.2}$ | $40.34_{1.3}$ |
| + Temporal | $51.69_{1.0}$ | $43.33_{1.5}$ | $39.24_{0.3}$ | $40.12_{0.7}$ | $39.50_{0.4}$ |
| + LMLM | $51.91_{0.9}$ | $43.26_{1.1}$ | $39.41_{1.2}$ | $40.94_{0.6}$ | $40.79_{0.5}$ |
| + STLM † | $52.48_{0.6}$ | $\mathbf{43.67_{0.2}}$ | $39.60_{0.8}$ | $41.21_{1.1}$ | $40.52_{0.9}$ |
| + SG-TLM ‡ | $\mathbf{52.89_{0.5}}$ | $43.51_{0.4}$ | $\mathbf{41.11_{0.7}}$ | $\mathbf{41.25_{0.5}}$ | $\mathbf{40.87_{0.6}}$ |

Table 2: Results of the **RTC** dataset: memorization and generalization performance on the **PSP** task (average of 5 runs). We utilize the set in *20-01* and *20-02* to test generalization and the rest to test memorization. Our proposed SG-TLM achieves the highest F1-score.

| **LMLM** (Su et al., 2022) | |
|---|---|
| The speed of representation | 200min $(1.0\times)$ |
| The speed of measuring | 360min $(1.0\times)$ |
| F1-score on average | 65.22 (−) |
| **SG-TLM** (Ours) | |
| The speed of parsing | 36min $(5.5\times)$ |
| The speed of measuring | 2min $(180\times)$ |
| F1-score on average | **66.67** (+1.45) |

Table 3: Efficiency comparison between LMLM and SG-TLM. SG-TLM is a faster and more effective model.

However, the performance differences among the methods in the **RTC** dataset are relatively minor compared to the **WMT** dataset, which can be attributed to the shorter time intervals and the relatively stable and slight dynamic temporality of the **RTC** dataset. Moreover, SG-TLM also shows a notable drop in future years due to the inherent uncertainty of future language changes (Lazaridou et al., 2021) in both datasets. In Section 4.5, we will investigate the approaches to refreshing the models as new data arrives and compare the temporal adaptation (Röttger and Pierrehumbert, 2021; Luu et al., 2022) performance of different methods.

### 4.4 Detailed Analysis

**Efficiency Comparison**  Table 3 shows the comparison of lexicon construction efficiency between LMLM and SG-TLM. LMLM provides a fairly complex and time-consuming method to select semantic-based lexicons, i.e., 200 minutes for representing and 360 minutes for clustering. However, SG-TLM benefits from the speed superiority of syntactic parsing tools rather than large-scale PLMs, resulting in a speedup ratio of $5.5\times$ compared to representation and $180\times$ compared to measuring. Furthermore, SG-TLM outperforms LMLM on the **TWINER** task by achieving a 1.45 higher F1-score, demonstrating its superior effectiveness.

**Ablation Study**  To investigate the impacts of different components within SG-TLM, we remove individual components from the complete model and observe the resulting performance. The results are shown in Table 4. Notably, excluding the SGM objective leads to the most significant decline in performance, highlighting its pivotal role within the SG-TLM framework. Furthermore, each component contributes positively to the overall performance, indicating the utility and significant contributions of all SG-TLM components in improving the model's effectiveness.

**Scale Effects in Performance**  We also explore whether our proposed SG-TLM would keep effective over random masking when increasing the amount of data. Using the **WMT** dataset, we successively expand the training data for both models,

| Model | PoliAff | | | | | | | TwiNER | | | | |
|---|---|---|---|---|---|---|---|---|---|---|---|---|
| | 2015 | 2016 | 2017 | 2018 | *2019* | *2020* | Avg. | 2016 | 2017 | 2018 | *2019* | Avg. |
| Our model | **93.69** | **76.69** | **66.58** | **55.94** | **57.41** | **50.32** | **66.77** | 81.78 | **61.85** | **61.18** | **61.87** | **66.67** |
| $\theta$ - SMLM | 92.66 | 75.08 | 65.35 | 53.14 | 55.61 | 48.10 | 64.98 | **82.71** | 59.25 | 60.52 | 59.56 | 65.51 |
| $\theta$ - TMLM | 92.86 | 76.75 | 66.02 | 55.48 | 57.12 | 49.20 | 66.20 | 78.62 | 60.37 | 60.42 | 60.37 | 64.95 |

Table 4: Ablation study: Results on the **PoliAff** and **TwiNER** task (average of 5 runs). We can discover that removing SGM from the SG-TLM has the most significant impact.

| Methods | TwiNER | | | | |
|---|---|---|---|---|---|
| | 2016 | 2017 | 2018 | *2019* | Avg. |
| Uniform$_{4M}$ | 77.04 | 59.05 | 58.52 | 59.07 | 63.42 |
| SG-TLM$_{4M}$ | **81.78** | **61.85** | **61.18** | **61.87** | **66.67** |
| Uniform$_{8M}$ | 77.18 | 56.78 | 55.21 | 56.8 | 61.49 |
| SG-TLM$_{8M}$ | **83.31** | **61.71** | **60.96** | **61.72** | **66.93** |
| Uniform$_{12M}$ | 79.12 | 59.12 | 58.03 | 59.32 | 63.89 |
| SG-TLM$_{12M}$ | **83.74** | **61.63** | **60.11** | **61.94** | **66.86** |
| Uniform$_{16M}$ | 81.31 | 59.94 | 60.38 | 58.96 | 65.14 |
| SG-TLM$_{16M}$ | **84.66** | **62.57** | **61.57** | **61.58** | **67.60** |
| Uniform$_{20M}$ | 80.47 | 60.01 | 59.40 | 60.41 | 65.07 |
| SG-TLM$_{20M}$ | **85.27** | **62.17** | **61.64** | **62.18** | **67.82** |

Table 5: Results of scale effects in performance (average of 5 runs). In method names, "4M", "8M", etc., denote the use of 4 million, 8 million, etc., datasets for pre-training. SG-TLM and Uniform represent our proposed method and the random masking strategy, respectively.

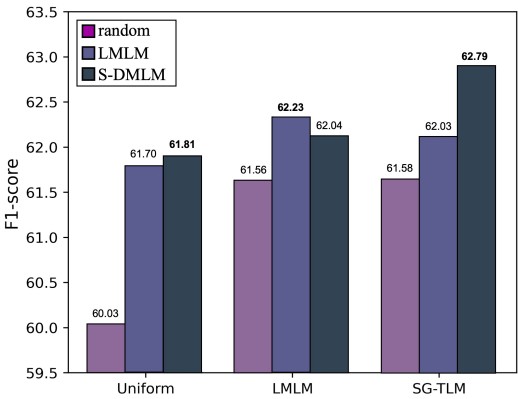

Figure 4: Results of different TLMs' adaptability performance on the target year (average of 5 runs). The horizontal axis indicates the different TLMs, and the vertical axis shows the F1-score of the **TwiNER** task. Our proposed SG-TLM outperforms other models with the highest F1-score among the nine settings.

multiplying the volume by factors of 2, 3, 4, and 5. The results of this scaling experiment are summarized in Table 5. It is clear that the SG-TLM consistently outperforms the uniform masking approach across all years and data scales, demonstrating the robustness of our approach with increasing data.

**Hyper-Parameter Analysis** Considering the correlation between the masking ratio and the model's performance, we conduct experiments to explore the most suitable masking ratio $a$ and word counts $k$ for the SG-TLM objective. The results are shown in Figure 5. Our SG-TLM achieves the best performance when the masking ratio $a$ is set to 30% and the number of candidate words $k$ is 200. To better understand the insights of our presented SG-TLM, we conduct token-level analysis on the selected lexicons in Appendix D.

### 4.5 Temporal Adaptation to New Data

Unlike domain adaptation, temporal adaptation (Röttger and Pierrehumbert, 2021) updates models with current data to mitigate temporal mis-

alignment[14]. In this section, we consider the scenario where we already have a trained model on the 2015-18 slices and new data from the 2019 slice. We attempt to update the model by continuing pre-training TLMs on the unlabelled target year data. To compare the adaptability of different TLMs on target year, we evaluate their performance on the **TwiNER** dataset. Precisely, we fine-tune the adaptation models on the labeled source year data[15] and then test models on 2019 data. Experiments are conducted adapting three TLMs with three lexicon construction methods, totaling nine settings. Results are shown in Figure 4.

### 4.6 Transferability Across PLM Frameworks

To verify the transferability of our methods across different model frameworks, we implement our method in both encoder-only and decoder-only models and utilize random lexicon construction as the baseline for comparison.

---

[14]In Appedix E, we will provide further analysis on SG-TLM's adaptability to temporal changes from the perspective of token prediction.

[15]In previous experiments, we use 2015 as the source year.

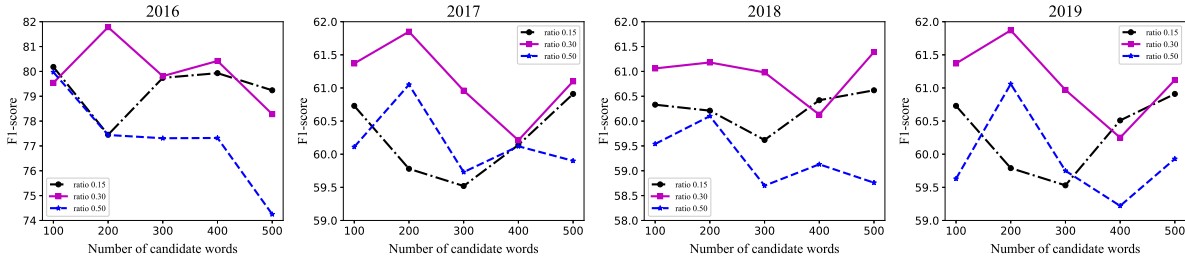

Figure 5: Results of the different masking strategies of SG-TLM (average of 5 runs). The horizontal axis indicates the candidate word counts $k$ and the vertical axis represents the F1-score of the **TwiNER** task.

| Model | RTC | | | | |
|---|---|---|---|---|---|
| | **17-04** | **18-04** | **19-08** | *20-01* | *20-02* |
| BERT | 51.97 | 42.57 | 39.98 | 40.75 | 40.53 |
| + SG-TLM | **52.89** | **43.51** | **41.11** | **41.25** | **40.87** |
| RoBERTa | 52.09 | 42.96 | 39.22 | 40.81 | 40.64 |
| + SG-TLM | **54.01** | **44.81** | **40.11** | **41.44** | **41.84** |

Table 6: Results of different PLMs on the **RTC** task (average of 5 runs). Our proposed SG-TLM achieves the highest F1-score on both BERT and RoBERTa.

**Effectiveness on Encoder-only PLMs** We implement our method on two popular encoder-only PLMs, i.e., BERT and RoBERTa. As shown in Table 6, we observe a significant improvement in each PLM using our SG-TLM. On average, the BERT model improves by 0.77 points, and the RoBERTa model improves by 1.31 points. These findings illustrate the versatility of our proposed SG-TLM, as it enhances the performance of various PLMs by incorporating syntax information.

**Effectiveness on Decoder-only PLM** We also conduct experiments on large-scale decoder-only language models, i.e., LLaMA-7B (Touvron et al., 2023). In these experiments, we extract the word selection component from our method and compare two data selection methods: one based on our **Sy**ntax-**G**uided (SyG.) approach and the other on random selection. The perplexity of LLaMA-7b is evaluated on the 2,000 sentences selected using these two methods. As shown in Table 7, SG-TLM yields higher perplexity than random selection in the **RTC** datasets[16]. This highlights the complexity and diversity of our selected data, indicating the effectiveness of incorporating syntax into data selection and the potential to enhance temporal capture in large language models.

---

[16]Since the training data for LLaMA is cut off at 2022, we crawl the latest Reddit data from `https://files.pushshift.io` to complete the experiment.

| Method | RTC | | | | |
|---|---|---|---|---|---|
| | **21-02** | **21-08** | **22-02** | **22-08** | **23-02** |
| Random | 14.81 | 14.98 | 15.02 | 15.29 | 15.46 |
| SyG. | **15.82** | **16.02** | **16.37** | **16.55** | **16.90** |

Table 7: Comparative perplexity results of LLaMA-7B on the **RTC** dataset using **Sy**ntactic-**G**uided (SyG.) and random selection methods.

## 5   Related work

**Temporal Language Model** Several works have explored the temporal effects in language models (Huang and Paul, 2018, 2019; Rijhwani and Preoţiuc-Pietro, 2020; Lazaridou et al., 2021; Søgaard et al., 2021; Agarwal and Nenkova, 2022; Loureiro et al., 2022; Cao and Wang, 2022; Cheang et al., 2023). Recently, existing works have investigated the temporal language model to model temporality information and generalize over time. Dhingra et al. (2022) and Rosin et al. (2022) directly prefix the time token to text sequences and fine-tune on time-stratified data. Hofmann et al. (2021) and Rosin and Radinsky (2022) modify the structure of the language model to create time-specific contextualized word representations. Su et al. (2022) recently proposed a semantic-based, lexical masking strategy to enhance PLMs' temporal generalization. We extend this work with a detailed study using syntactic role changes to harness temporal-specific information efficiently.

**Diachronic Language Change** Diachronic language change can be mainly divided into semantic change (Kurtyigit et al., 2021; Montanelli and Periti, 2023), morphological change (Hare and Elman, 1995; Ji et al., 2019; Giulianelli et al., 2022), and syntactic change (Kroch, 2001; Seretan, 2011; Bybee, 2017; Merrill et al., 2019). Previous work focused on discovering the words that have undergone diachronic change under the supervised

settings (Kim et al., 2014; Basile et al.; Basile and McGillivray, 2018; Tsakalidis et al., 2019; Kurtyigit et al., 2021). Recently, several works have demonstrated that contextualized word representations have dynamic representation capabilities (Pilehvar and Camacho-Collados, 2019; Chronis and Erk, 2020; Garí Soler and Apidianaki, 2021; Laicher et al., 2021; Qiu and Xu, 2022), which are adopted with unsupervised methods to represent, cluster, and differentiate words across different time periods (Giulianelli et al., 2020; Montariol et al., 2021). Our method utilizes a syntax-based method to detect the salient change words within the text sequence, making the process more interpretable (Merrill et al., 2019; Ryzhova et al.; Kutuzov et al., 2021). To our best knowledge, this is the first work that incorporates syntactic knowledge into the training of temporal language models.

## 6 Conclusion

In this paper, we enhance the temporal language model from the syntactic perspective and discover that predicting the syntax-guided lexicons is more challenging for static PLMs compared to other methods. Building upon these findings, we propose a syntax-guided temporal language model (SG-TLM) that incorporates time information into tokens with significant syntactic changes. Our SG-TLM achieves the SOTA performance, reduces computational costs during lexicon construction, and demonstrates excellent transferability to new data and frameworks compared to other baselines.

## 7 Limitation

There are still some limitations in our work which are listed below:

- While we introduce a data selection strategy that incorporates syntactic changes to identify time-specific sentences, we only conduct preliminary validation of our method's transferability on Large Language Models (LLMs), without involving training and inference stages. Recent studies highlight that LLMs continue to struggle with effective generalization when it comes to emerging data (Wang et al., 2023). As a potential solution, our future work aims to integrate our method with in-context learning (Dong et al., 2022) to enhance the temporal generalization capabilities of LLMs.
- Recent studies (Kutuzov et al., 2021; Giulianelli et al., 2022) show the effectiveness of utilizing

syntactic features in detecting lexical semantic changes. This prompts us to investigate the compatibility of our method with the semantic lexicon solution. Given the lexicon constructed by semantic change $\mathcal{W}_l$ and syntactic change $\mathcal{W}_s$, we conduct three straightforward methods to combine the lexicons between $\mathcal{W}_l$ and $\mathcal{W}_s$: $\mathcal{W}_l \cap \mathcal{W}_s$, $\mathcal{W}_l \cup \mathcal{W}_s$, $\mathcal{W}_l \setminus \mathcal{W}_s$. Consistent with the previous experiment, the masking ratio $a$ is 30%, and $k$ is 200. The results are shown in Table 8. Contrary to our expectations, the combined model's performance is inferior as compared to the original model, suggesting that this method fails to merge information effectively from multiple dimensions. In future work, we will explore more suitable methods to integrate semantic and syntactic information.

| Model | RTC | | | | |
|---|---|---|---|---|---|
| | 17-04 | 18-04 | 19-08 | 20-01 | 20-02 |
| $\mathcal{W}_l$ | 51.91 | 43.26 | 39.41 | 40.94 | 40.79 |
| $\mathcal{W}_s$ | **52.89** | **43.51** | **41.11** | **41.25** | **40.87** |
| $\mathcal{W}_l \cap \mathcal{W}_s$ | 50.20 | 42.25 | 38.60 | 39.54 | 39.65 |
| $\mathcal{W}_l \cup \mathcal{W}_s$ | 50.34 | 42.32 | 38.28 | 40.12 | 39.99 |
| $\mathcal{W}_l \setminus \mathcal{W}_s$ | 50.24 | 42.33 | 38.66 | 40.13 | 39.86 |

Table 8: The compatibility of our method with semantic lexicon solution. $\mathcal{W}_l$ represents lexical semantic solution, while $\mathcal{W}_s$ represents our syntactic role solution.

## Acknowledgement

We would like to thank the anonymous reviewers for their constructive comments. This work was supported by the National Science Foundation of China (NSFC No. 62206194) and the Natural Science Foundation of Jiangsu Province, China (Grant No. BK20220488).

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

## A Implementation of Experimental Setup

### A.1 Task Description

The task of political affiliation classification (**POLIAFF**) revolves around determining the political alignment of individuals based on a given text. The model is fine-tuned to categorize the content into Republican or Democrat, providing insights into the individual's political leaning. In contrast, named entity recognition (**TWINER**) focuses on the identification and classification of named entities within a text. The model is tasked with recognizing and categorizing text into three distinct entity types: people (*PER*), locations (*LOC*), and organizations (*ORG*). Furthermore, political subreddit prediction (**PSP**), derived from the Reddit Time Corpus (**RTC**) dataset, is a five-way classification task. The goal is to categorize a provided text into one of five political subreddits: *r/donald*, *r/libertarian*, *r/conservative*, *r/politics*, and *r/chapotraphouse*.

### A.2 Model Training & Hyper-Parameters

**Architecture** For all our experiments, we utilize the Hugging-Face transformer package[17] to implement language models. Our architecture is the BERT-base uncased model, based on a large corpus of English data with MLM objective for pretraining. The model includes 12 transformer layers and 12 attention heads; the hidden layer size is 768, and the parameter number is 110 million in total. For the downstream classification task, we add a linear layer after the last BERT layer. The output is generated using softmax. The maximum input sequence length that the model can handle is 128.

**Training details** During the pre-training stage, we set the candidate word counts $k$ to 200 and the masking ratio $a$ to 30%. We analyze different masking strategies in Appendix 4.4. In both the pre-training and fine-tuning stages, we employ the cross-entropy loss as the objective function and utilize the AdamW (Loshchilov and Hutter, 2017) optimizer. The learning rate is set to 5e-5, and the weight decay is set to 0.01. To prevent overfitting, we apply a dropout probability of 10% for regularization. The model is pre-trained for five epochs and fine-tuned until convergence. We use a batch size of 128 and conduct the experiments on eight NVIDIA A5000 GPUs.

---

[17]https://huggingface.co

**Evaluation Metric** We utilize the F1 score[18] as the evaluation metric in all the experiments.

## B Implementation of the Baselines

This section further provides experimental details of several TLMs used as baselines in Section 4.2: **TAda**, **Temporal**, and **LMLM**. Following the strategy by (Su et al., 2022), **LMLM** utilizes 500 candidate words and a mask ratio of 0.3. For the remaining baselines, the mask ratio aligns with the standard BERT (Devlin et al., 2019) configuration at 0.15.

**TAda** The TAda (Röttger and Pierrehumbert, 2021) approach underlines the importance of reflecting temporal dynamics by training distinct models, each tailored to specific time intervals. To this end, we initiate an ensemble of models, each trained on unlabeled data corresponding to a specific year. Each model, or "yearly expert," thus becomes specialized to the linguistic nuances of its respective time bucket. During testing, a test input is provided to the model that matches its timestamp. If the timestamp of the test data falls outside the range of the training data, we choose the model trained on the closest year to make predictions. This setup enables the model to adapt to temporal changes by dedicating separate models to different time periods.

**Temporal** On the other hand, the Temporal (Dhingra et al., 2022) baseline is trained on the entire dataset as a single model. The unique feature of this setup is the way it incorporates time into the input. Specifically, the model takes in a concatenation of the time $t$ and the input $x$, that is, $P(y|x, t; \theta) = P(y|t \oplus x; \theta)$. This is achieved by prefixing the input with a simple string representation of time, such as "year: 2014". Thus, the model is trained to generate outputs based on both the input and its associated time, allowing it to develop a sense of temporal sensitivity.

**LMLM** In the lexical-based Masked Language Model (LMLM) (Su et al., 2022) setup, the model is trained to account for semantic shifts in words over time. This is achieved by constructing a lexicon of words that have exhibited significant semantic changes over time and then using this lexicon in a masked language model setup during pre-training.

---

[18]https://scikit-learn.org/stable/modules/generated/sklearn.metrics.f1_score.html

| Methods | TWiNER | | | | |
|---|---|---|---|---|---|
| | 2016 | 2017 | 2018 | *2019* | Avg. |
| Stanza | **81.78** | **61.85** | **61.18** | **68.27** | **68.27** |
| UDPipe | 78.50 | 60.59 | 59.41 | 60.60 | 64.78 |

Table 9: Results of different syntactic methods under the time-stratified settings (average of 5 runs).

| Cos. ($\uparrow$) | 2015 | 2016 | 2017 | 2018 | SyC. |
|---|---|---|---|---|---|
| 0.00~0.01 | 328 | 342 | 251 | 298 | micro |
| 0.01~0.02 | 115 | 89 | 139 | 123 | medium |
| 0.02~0.03 | 35 | 28 | 44 | 35 | great |
| 0.03~1 | 22 | 41 | 66 | 44 | great |

Table 10: Distribution of the syntactic changed words, where SyC. represents for the **S**yntactic **C**hange.

| Year | ADJ | ADV | INTJ | NOUN | NUM | PROPN | VERB |
|---|---|---|---|---|---|---|---|
| 2015 | 1,073 | 19 | 0 | 18,690 | 0 | 74,074 | 3,654 |
| 2016 | 1,715 | 0 | 0 | 26,718 | 12 | 68,690 | 6,887 |
| 2017 | 817 | 11 | 0 | 41,173 | 0 | 70,919 | 4,651 |
| 2018 | 673 | 493 | 1 | 25,688 | 0 | 56,576 | 4,171 |

Table 11: The distribution of the selected lexicons by Part of Speech (POS) across various years in the WMT pre-trained dataset.

The model is thus trained to predict the original word based on the context and the time token, allowing it to capture temporal dynamics in word semantics.

For all these baselines, we perform fine-tuning on the same downstream task data, ensuring fair comparisons among the models. Moreover, we use the same `BERT-BASE-UNCASED` pre-trained model as the foundational language model for all baselines, setting a level playing field.

As a variant of **LMLM** and **SG-TLM**, we also consider an additional baseline approach where timestamps are treated as a prefix at the beginning of the sentence and are randomly replaced with the `<MASK>` token for prediction. Meanwhile, the remaining tokens within the sentence are randomly masked. This baseline is equivalent to employing Temporal-Aware Masking (TAM) with random token masking in sentences, corresponding to the experiment with $\theta$ - SMLM presented in Section 4.4. Notably, despite these different masking conditions, our proposed method SG-TLM consistently outperforms the other models.

## C  Parsing Toolkit Analysis

Since there is a strong correlation between the parsing toolkit's capability and the performance of our SG-TLM, we compare the selected Stanza with another commonly adopted parsing tool, i.e., UDPipe[19]. As shown in Table 9, the results of utilizing Stanza as the parsing method outperforms UDPipe in all the timestamps, i.e., the average accuracy of Stanza is 68.27, while UDPipe is 64.78, which indicates that utilizing a more accurate parsing tool can significantly improve the model's performance. Though the UDPipe does not depend on GPU resources, this toolkit is unsuitable for parsing syntactic roles in the selection process of lexicons.

---

[19]UDPipe (Straka and Straková, 2017) utilizes fast transition-based neural dependency parser that follows the same annotation schemes as Stanza.

## D  Token-level Analysis

From Figure 5, it is surprising that there is no positive correlation between the word counts and the model's performance. To understand the reason behind this phenomenon, we select the top 500 words from the candidates $\mathcal{W}_{mask}^t$ according to the cosine value. The distribution of those words is shown in Table 10, which indicates that only about 20% words have relatively significant syntactic change (cosine value $\geq 0.01$). This suggests that performance mainly comes from correctly predicting a small number of keywords, such as topic words and newly emerging words, which have relatively salient syntactic change. Furthermore, we also show the distribution of these lexicons by Part of Speech (POS) across various years in Table 11. From the data, it's evident that the distribution of POS remains relatively consistent year-over-year. Nouns dominate the distribution, implying their higher propensity for syntactic variation.

## E  Superiority in Adapting Temporal Change

This section aims to demonstrate the superior performance of the SG-TLM model in adapting to temporal changes compared to other Temporal Language Models (TLMs). Specifically, we compare the SG-TLM model against established baselines, i.e., Uniform, Temporal, and LMLM, as previously introduced in Section 4.2. All models are further pre-trained on BERT using 4 million data instances spanning from 2015 to 2018 in the WMT dataset and evaluated to predict masked tokens at different timestamps using 2,000 data samples from the

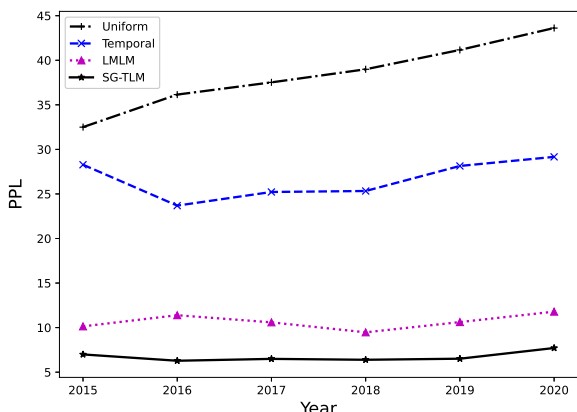

Figure 6: Results of adapting to the temporal shift. The SG-TLM consistently achieves the lowest *ppl.* scores compared to Uniform, Temporal, and LMLM models from 2015 to 2020.

same source spanning six years (2015-2020). This experimental setting has also been used to verify the ability to adapt to temporal changes in Röttger and Pierrehumbert (2021). The results of this comparison are presented in Figure 6. We can observe that the SG-TLM consistently outperforms the other models across all examined years, achieving the lowest perplexity scores, which illustrates the superior adaptability of the SG-TLM model to temporal shifts. Notably, the SG-TLM model also exhibits superior generalization capability with respect to the 2019 and 2020 data that were not present during the training period, outperforming other baseline models, which further demonstrates its robustness and reliability in handling temporal shifts.