# OpenReview forum: "Efficient Continue Training of Temporal Language Model with Structural Information"
_EMNLP/2023/Conference — EMNLP 2023 Findings_

### Official Review · Reviewer_shTU · 2023-08-04

**Soundness:** 4

**Excitement:**

3: Ambivalent: It has merits (e.g., it reports state-of-the-art results, the idea is nice), but there are key weaknesses (e.g., it describes incremental work), and it can significantly benefit from another round of revision. However, I won't object to accepting it if my co-reviewers champion it.

**Paper Topic And Main Contributions:**

This paper aims to improve the efficiency of continue training on Temporal PLMs. It launches a thorough study to explore the effects of different methods for lexicon selection and proposes a simple syntax-guided temporal language model to capture time information in syntactically changed tokens. It achieves SOTA performance over baselines with a higher efficiency.

**Reasons To Accept:**

1.	Though simple, the idea of capturing temporal information via salient syntactic change distribution is novel to me. The competitive results demonstrate its validity while achieving a higher efficiency.
2.	This paper is well organized and pictures are clear to understand. Thorough experiments are conducted and fully demonstrated the effectiveness of SG-TLM and its each component.


**Reasons To Reject:**

1.	Since module “Syntax-Guided Lexicon Selection” needs to parse the syntax for each sentence, it depends heavily on the accuracy of Parsing Toolkit, which can be a major influence on overall F1 score.
2.	As mentioned in section 2.2, author conducts a strong assumption that temporal information can be captured in the changes of syntactic role across time. But it neglects the situation that some words’ semantic information change while their syntactic role remains, thus temporal information cannot reflect accurately from the syntactic role distribution. As mentioned in limitations, both lexical semantic and syntactic information need to be integrated.
3.	In section 2.2, though the SMLM makes sense to me, the way to convert syntactic dictionaries to feature vectors is not clear(What do two vectors a_t exactly indicate?)

**Reproducibility:**

4: Could mostly reproduce the results, but there may be some variation because of sample variance or minor variations in their interpretation of the protocol or method.

**Reviewer Confidence:**

4: Quite sure. I tried to check the important points carefully. It's unlikely, though conceivable, that I missed something that should affect my ratings.

---

> ### Author Rebuttal · Authors · 2023-08-29
>
> Dear Reviewer shTU,
>
> Thank you for the insightful comments and suggestions. We value the input and would like to address your concerns as follows:
>
> **Dependency on Parsing Toolkit**: We acknowledge that the "Syntax-Guided Lexicon Selection" module's performance depends on the accuracy of the Parsing Toolkit. However, it's essential to note that modern parsers are highly **effective, efficient and easily accessible**, making them **reliable** for many NLP tasks. We also perform preliminary experiments with different parsers to ensure we use the most accurate tool in Appendix C.
>
> **Balancing Syntactic and Semantic Dimensions in Temporal Information Capturing**: It is a nice suggestion! As you pointed out, there exists scenarios where the semantic information of certain words undergoes a change, while their syntactic roles remain constant.
>
> To tackle the challenge you've highlighted, we conducted basic experiments, as detailed in the limitations section, to combine syntactic and semantic aspects. Admittedly, these initial attempts did not yield the desired integration.
>
> This is an unresolved yet compelling issue that requires further research. We will next explore the interaction from multiple dimensions of linguistic attributes and investigate which type of information has  advantages in modeling language evolution.
>
> **Clarity on SMLM in Section 2.2:**
>
> Thank you for pointing out the need for clarity in Section 2.2. Here is a detailed explanation:
>
> - **Identification of Unified Set of Keys**: We start with two syntactic dictionaries, $R_1$ and $R_2$, corresponding to different time periods. A unified set of keys, denoted as $S$, is formed by taking the union of keys from both dictionaries: $S = \text{keys}(R_1) \cup \text{keys}(R_2)$.
> - **Frequency-based Refinement**: We then refine $S$ based on a frequency threshold. Only keys with frequencies exceeding 5% of the total number of keys in $S$, resulting in a refined set $S’$: $S' = | k \in S: \text{frequency}(k) > 0.05 \times \text{total number of keys in } S |$.
> - **Alignment of Dictionaries**: The dictionaries $R_1$ and $R_2$ are adjusted to align with $S’$. For each key in $S’$, if it exists in $R_1$ and $R_2$, its value is retained. Otherwise, it's assigned a value of zero: $R'_1[k] = R_1[k] \, \text{if } k \in R_1 \, \text{else } 0; R'_2[k] = R_2[k] \, \text{if } k \in R_2 \, \text{else } 0$.
> - **Vector Formation**: Finally, vectors $a_1$ and $a_2$ are formed using the values from the aligned dictionaries $R_1’$ and $R_2’$: $a_1 = \text{values}(R'_1); a_2 = \text{values}(R'_2)$.
>
> Finally, we want to thank the reviewers again for their valuable feedback. Hope that the above explanations address your concerns.

---

### Official Review · Reviewer_PDRu · 2023-08-04

**Soundness:** 4

**Excitement:**

4: Strong: This paper deepens the understanding of some phenomenon or lowers the barriers to an existing research direction.

**Missing References:**

None.

**Paper Topic And Main Contributions:**

The authors target the static nature of current Pre-trained Language Models, most of which are trained on snapshots of language data with no temporal information. Building on existing work which simply adds timestamps, the authors identify that existing PLMs exhibit structural biases present in their training data. Their remedy proposes to use this syntactic bias to direct masking while pretraining, while concurrently adding temporal masking. These lead to noticeable improvements over existing models, both in terms of memorization and generalization.

The major contributions of this work are:
- identifying generalizability weaknesses of current PLMs due to syntactic role biases for lexical items in the dataset.
- proposing a novel method for lexicon selection that chooses a set of lexical items that change substantially in syntactic role over a period of time.
- Pre-training an MLM with targeted masking of words in their lexicon, as well as temporal masking, and showcasing the improvements of such a PLM on relevant tasks.

**Questions For The Authors:**

Consider points 2 and 3 in Reasons to reject - I put in a lot of questions there.

**Reasons To Accept:**

1. The paper's aim, and proposed solution are clear and I believe novel. Furthermore, the authors implement the syntax guided temporal masking solution as detailed, and show its advantages over existing solutions. While highly specific towards syntactic biases in data, I can imagine this technique being more generalizable towards countering other biases.

**Reasons To Reject:**

1. The results and the interpretations would be bolstered with statistical tests, or at the very least, multiple runs with means and S.D listed. I understand that these are huge models and each fine-tuning experiment might take hours. It is just harder to interpret these numbers as being intrinsically better when they are so close to one another, like in Figure 4.
2. Table 10 and its related discussion is a little worrying. Firstly, I think some preliminary analysis of the actual selected lexicon should be in the main paper and not the appendix. What is the distribution of selected lexicon changes in the pretraining data? How many nouns, verbs etc. Which are the most common role changes?..There are many questions there of course. Furthermore as Appendix E states, most of the syntactic change is from a few keywords. Is this PLM then modeling language change as a general category, or idiosyncratic key word spikes or losses that happen due to news events? The latter are a subset of the former, and perhaps might be one reason for lower generalizability results (even with the SG-TLM model being the best).
3. Continuing from my previous point, I am worried that the method for identifying structural biases is not very generalizable, and is dataset specific. For instance, the datasets used in the paper are political datasets. The word 'Oval' I would hazard changes roles leading up to a US election, or in turbulent administrations like Trump (which might explain the biases shown in Figure 1). I feel calling this `language change` or `linguistic drift` is too strong a claim. It implies that people who the dataset represents, genuinely evolved in their usage of that lexicon. But is that actually the case? It rather seems to capture the fact that certain real-world events lead to some distributional changes in language use, but not a shift or change in the word itself.

It might seem a trivial point, but Table 10 does lead me to think that perhaps using another phrase from 'language change', as well as orienting the pitch of the paper a little differently might help counter this perception that I get.

**Reproducibility:**

4: Could mostly reproduce the results, but there may be some variation because of sample variance or minor variations in their interpretation of the protocol or method.

**Reviewer Confidence:**

4: Quite sure. I tried to check the important points carefully. It's unlikely, though conceivable, that I missed something that should affect my ratings.

**Typos Grammar Style And Presentation Improvements:**

Figure 1 is tiny and compressed, should be bigger in camera ready. Also, mention the data you extracted the counts from when referring to Figure 1, in caption and perhaps Line 81. The numbers in all the charts need to be set in a larger font as well.

---

> ### Author Rebuttal · Authors · 2023-08-29
>
> Dear Reviewer PDRu,
>
> Thank you for your detailed feedback and insights on our paper. Below, we have organized your concerns and our corresponding answers:
>
> > Concern 1: Results and interpretations would benefit from enhanced statistical validation.
>
> Answer1: **All the results presented in the paper are derived from an average taken over five random seeds**. Here are the results on the **RTC** dataset with the annotated standard deviations (S.D):
>
> | **Model**  | **17-04**  | **18-04**  | **19-08**  | _**20-01**_ | _**20-02**_ |
> | ---------- | ---------- | ---------- | ---------- | ----------- | ----------- |
> | BERT       | 49.61 ±0.8 | 41.74 ±1.1 | 38.23 ±0.5 | 38.84 ±1.2  | 39.97 ±0.3  |
> | + Uniform  | 51.97 ±0.7 | 42.57 ±1.3 | 39.98 ±0.4 | 40.75 ±0.9  | 40.53 ±1.0  |
> | + TAda     | 50.62 ±0.6 | 42.21 ±0.9 | 38.43 ±1.4 | 39.47 ±0.2  | 40.34 ±1.3  |
> | + Temporal | 51.69 ±1.0 | 43.33 ±1.5 | 39.24 ±0.3 | 40.12 ±0.7  | 39.50 ±0.4  |
> | + LMLM     | 51.91 ±0.9 | 43.26 ±1.1 | 39.41 ±1.2 | 40.94 ±0.6  | 40.79 ±0.5  |
> | + STLM†    | 52.48 ±0.6 | 43.67 ±0.2 | 39.60 ±0.8 | 41.21 ±1.1  | 40.52 ±0.9  |
> | + SG-TLM‡  | 52.89 ±0.5 | 43.51 ±0.4 | 41.11 ±0.7 | 41.25 ±0.5  | 40.87 ±0.6  |
>
> In forthcoming versions of the paper, we will include the S.D across these seeds to further validate the consistency and stability of our model.
>
> > Concern 2: What is the distribution of selected lexicon changes in the pre-training data?
>
> Answer2: We recognize the importance of a detailed analysis of the selected lexicon and initially placed it in the appendix for readability. However, based on your feedback, we've now included this deeper analysis in the main paper. The table below shows t**he distribution of these lexicons by Part of Speech (POS)** across various years in the WMT pre-trained dataset:
>
> | Year | ADJ   | ADV  | INTJ | NOUN   | NUM  | PROPN  | VERB  |
> | ---- | ----- | ---- | ---- | ------ | ---- | ------ | ----- |
> | 2015 | 1,073 | 19   | 0    | 18,690 | 0    | 74,074 | 3,654 |
> | 2016 | 1,715 | 0    | 0    | 26,718 | 12   | 68,690 | 6,887 |
> | 2017 | 817   | 11   | 0    | 41,173 | 0    | 70,919 | 4,651 |
> | 2018 | 673   | 493  | 1    | 25,688 | 0    | 56,576 | 4,171 |
>
> From the data, it's evident that the distribution of POS remains **relatively consistent** year-over-year. **Nouns dominate the distribution**, implying their higher propensity for syntactic variation. We will incorporate more detailed analysis into the main paper in the next version.
>
> > Concern 3: Is this PLM then modeling language change as a general category, or idiosyncratic key word spikes or losses that happen due to news events?
>
> Answer3: It’s a good question! **New events**, like `the US election`, do cause temporary vocabulary shifts. However, such shifts, though vivid, are typically **short-lived**; when these events are no longer the focal point, language usage will revert to its prior state [1]. Meanwhile, these fluctuations are detectable via straightforward statistical patterns [2].
>
> However, **our study's focus is on the deeper, lasting shifts in language, not just short-term reactions to events.** As illustrated in Figure 2, such changes are challenging to capture purely based on statistical frequency, yet they significantly affect a model's comprehension capability over time.
>
> To further buttress our claim, we **trace the lexicon evolution** over a span from 2015 to 2020, a duration long enough to transcend the effects of individual events. Within these lexicons, out of all the words selected from each year, **23% showed consistent trends of change**, which validates our primary assumption: our model observing shifts emblematic of broader, long-term linguistic evolution, rather than just transient, event-induced fluctuations.
>
> > Concern 4: Is the method for identifying biases specific to political datasets and not generalizable?
>
> Answer4:
>
> - **Diversity in Validation Settings:** It's crucial to underscore that our research **isn't solely rooted in political datasets.** In fact, the majority of our experiments, as highlighted in Table 1, 4, 5, and 9, are conducted on the WMT dataset. This dataset offers **a wide array of news sources** that span various topics, domains, and styles, extending beyond just politics. It’s worth noting that our chosen datasets are **standard benchmarks** for studying temporal generalization and are **widely recognized** for evaluating a model's capability in capturing and understanding temporal shifts.
> - **Generalization on unseen data:** SG-TLM also performs well on unseen data. Our performance surpasses the baseline models (uniform vs. SG-TLM): **Poliaff** (2019: 55.75 vs. 57.41, 2020: 46.57 vs. 50.32), **TWINER** (2019: 59.07 vs. 61.87), **RTC** (2020-01: 40.75 vs. 41.25, 2020-02: 40.53 vs. 40.87). **It is worth noting that our model was tested on the brand new data it hasn't seen during training.** If our model was only good at remembering past specific language trends tied to events, it wouldn't have done well on this new data. This can also prove that our model understands broader language changes and don't just memorize past events.
>
> > Concern 5:  capture the fact that certain real-world events lead to some distributional changes in language use, but not a shift or change in the word itself.
>
> Answer5: As outlined in Answer 3, our study's focus is on the **deeper, lasting shifts** in language, substantiated by our experiments on lexical evolution (as seen in Answer 3) and data generalization (as seen in Answer 4). Therefore, we believe the terms "language change" or "linguistic drift" are not an overstatement.
>
> To further address your concerns, the tables below show two main ways `"oval"` has been used: as an 'amod' (adjectival modifier) and an 'nmod' (nominal modifier). By comparing **the counts and examples across years**, we can see trends and shifts in usage.
>
> | Syntax Label | 2015 | 2016 | 2017 | 2018 |
> | ------------ | ---- | ---- | ---- | ---- |
> | amod         | 82   | 142  | 264  | 232  |
> | nmod         | 192  | 75   | 60   | 46   |
>
> |      | **amod (adjectival modifier)**                               | **nmod (nominal modifier)**                                 |
> | ---- | ------------------------------------------------------------ | ----------------------------------------------------------- |
> | 2015 | She bought an **oval** table for her dining room.             | The beauty of the gem's **Oval** surpassed her expectations. |
> | 2016 | The artist released a series of paintings on **oval** canvases. | The window's **oval** has an intricate design.              |
> | 2017 | The designer released a watch with an **oval** face.         | Jewelers admire the brilliance of the diamond's **Oval**.   |
> | 2018 | She selected an **oval** frame for her portrait.             | The park's **oval** is used for various events.             |
>
> ​	Analysis reveals:
>
> - **Shift Towards Simplification**: We find a marked increase in the use of `"oval”` as `amod`. This is because using `"oval"` as an adjective straightforwardly describes an object's shape. **Language's shift towards simplicity** drives the increasing frequency of this usage, e.g. `She selected an oval frame for her portrait`.
> - **Decline in Nominal Modifier Usage**: The declining trend of `"oval"` being used as `nmod` might suggest its lesser practical application in daily communication. This can be viewed as a reflection of the natural evolution of language, uninfluenced by specific events.
>
> In conclusion, the changes in the usage of `"oval"` extend beyond specific events. Our qualitative analysis also shows that the pairing with `"oval office”` only **appeared 3 and 4 times** in 2016 and 2017 respectively even durning the US election. While we acknowledge that significant events can lead to linguistic shifts, our analysis suggests that the trends noted are more likely a reflection of the organic evolution of language.
>
> The size and font of Figure 1 have been adjusted accordingly.
>
> Again, thanks for your reviews, and we will definitely put these discussions in the new version. Hope the above answers can help address your concerns.
>
> [1] de Saussure, Ferdinand. *Course in general linguistics*. Columbia University Press, 2011.
>
> [2] Röttger P, Pierrehumbert J. Temporal adaptation of BERT and performance on downstream document classification: Insights from social media, EMNLP-21

---

### Official Review · Reviewer_uyuP · 2023-08-06

**Soundness:** 4

**Excitement:**

3: Ambivalent: It has merits (e.g., it reports state-of-the-art results, the idea is nice), but there are key weaknesses (e.g., it describes incremental work), and it can significantly benefit from another round of revision. However, I won't object to accepting it if my co-reviewers champion it.

**Paper Topic And Main Contributions:**

* The paper proposes a novel syntax-guided temporal language model (SG-TLM) with two masking strategies - syntax guided masking and temporal aware masking to incorporate temporal information in pertained language models.
* It introduces a syntax-guided lexicon selection method to identify words that have undergone significant syntactic changes over time and these words are masked during post-tuning stage.
* The proposed SG-TLM outperforms other baselines for incorporating temporal information in  language models on downstream tasks. It also demonstrates improved memorization and generalization abilities.
* Experiments show SG-TLM is more efficient for lexicon selection compared to semantic-based methods, with 5.5x faster parsing and 180x faster measurement.
* Analysis indicates SG-TLM consistently improves performance when transferred to different model frameworks (encoder-only, decoder-only models).

**Questions For The Authors:**

A:  The techniques seem to imply that the domain of the data/task and as a result the vocabulary remains fixed over time. It is not very clear to me how new words that are added over time would impact the performance of the proposed techniques. Ex. words like Moderna, covid-19 when considering sentiment analysis task related to vaccines over time.

**Reasons To Accept:**

* Very well written paper - clear and easy to read.
* Well thought out and intuitive motivation for proposed techniques to incorporate temporal information into language models.
* Experiments demonstrate that the techniques are transferable across model architectures which is encouraging.
* Efficiency of the technique is also a big plus.
* Strong empirical improvements over considered baselines.

**Reasons To Reject:**

* Additional qualitative analysis as to how/why the techniques lead to an improvement would be nice to include in the paper.

**Reproducibility:**

4: Could mostly reproduce the results, but there may be some variation because of sample variance or minor variations in their interpretation of the protocol or method.

**Reviewer Confidence:**

4: Quite sure. I tried to check the important points carefully. It's unlikely, though conceivable, that I missed something that should affect my ratings.

---

> ### Author Rebuttal · Authors · 2023-08-29
>
> Dear Reviewer uyuP,
>
> Thank you for your thoughtful review and comments on our paper. We provide the following answers for your concerns.
>
> > Question1: Additional qualitative analysis as to how/why the techniques lead to an improvement would be nice to include in the paper.
>
> Answer1: We appreciate your point on enriching the paper with a qualitative analysis.
>
> In response, we've used the running example `”Oval"` from Figure 1 throughout the paper to clarify how our techniques lead to improvement.
>
> Additionally, we extend our analysis to explore the `”Oval”` influence on downstream tasks. Take **TWINER** (named entity recognition) for example:
>
> sentence: `Oval Technologies announced their new product launch (2017)`
>
> label: `Oval Technologies`--`ORG`
>
> Uniform random masking fails to account for the diachronic change of `“Oval”`, leading to inaccurate entity recognition. However, our SG-TLM, informed by syntactic observations, prioritizes learning the semantic representation of `"Oval"`. The end result is its correct identification as an `ORG`.
>
> In next version of our paper, we plan to provide a more comprehensive qualitative analysis to further underline the effectiveness of our approach in handling syntactic shifts.
>
> > Question2: How new words that are added over time would impact the performance of the proposed techniques?
>
> Answer2: Our primary focus in this paper revolves around the distributional shifts caused by syntactic role change. The use of a fixed vocabulary in both our proposed model and the compared baselines is grounded in this focus.
>
> However, we acknowledge the potential impact of new words like "Moderna" or "covid-19”. Even though these words are new, their sentence patterns **will not dramatically change in the next few years**. Our model's primary strength lies in capturing these structural shifts, making it well-suited to understand and generalize the role of new words based on their contextual usage.
>
> Supporting this, our extensive experiments demonstrate the superiority of our approach in terms of **generalization to future datasets** (as seen in Table 1, Table 2, and Figure 6) and **adaptability to linguistic drifts** (evident from Figure 4).
>
> These findings highlight that even with a fixed vocabulary, our model's emphasis on structural changes equips it to understand and adjust to the impact of new terms.
>
> Hope the above replies can answer your questions.

---

### Meta-Review · Area_Chair_sLn1 · 2023-09-19

**Recommendation:** 3

**Metareview:**

Proposes a continued pretraining scheme that tries to incorporate temporal (date) information into representations in an effort to better generalize on down-stream tasks where the year is given in addition to input text. The method involves using off-the-shelf parsers to guide masking in BERT-like MLM. While performance appears to improve on selected downstream tasks (TwiNER and Poliaff) compared to baselines such as continued pretraining with prepending the date to the input text.

One concern that was raised was that the work depends on a well-working parser. This happens to work well for well written news, but may suffer for less well-formed text. It is unclear how general this can be beyond well-written text. Furthermore, a more comprehensive evaluation involving more tasks would be more convincing.

---

### Decision · Program_Chairs · 2023-10-07

**Decision:**

Accept-Findings

**Comment:**

Proposes a continued pretraining scheme that tries to incorporate temporal (date) information into representations in an effort to better generalize on down-stream tasks where the year is given in addition to input text. The method involves using off-the-shelf parsers to guide masking in BERT-like MLM. While performance appears to improve on selected downstream tasks (TwiNER and Poliaff) compared to baselines such as continued pretraining with prepending the date to the input text.

One concern that was raised was that the work depends on a well-working parser. This happens to work well for well written news, but may suffer for less well-formed text. It is unclear how general this can be beyond well-written text. Furthermore, a more comprehensive evaluation involving more tasks would be more convincing.